# Scale Effects on Shear Strength of Rough Rock Joints Caused by Normal Stress Conditions

**Jiayi Shen [1,2], Chenhao Sun [1], Huajie Huang [1], Jiawang Chen [3]** and **Chuangzhou Wu [1,2,]***

[1]    Institute of Port, Coastal and Offshore Engineering, Zhejiang University, Hangzhou 310015, China
[2]    Key Laboratory of Offshore Geotechnics and Material of Zhejiang Province, Zhejiang University, Hangzhou 310058, China
[3]    Hainan Institute, Zhejiang University, Sanya 572025, China
*    Correspondence: ark_wu@zju.edu.cn

**Abstract:** Scale effects on the mechanical behavior of rock joints have been extensively studied in rocks and rock-like materials. However, limited attention has been paid to understanding scale effects on the shear strength of rock joints in relation to normal stress $\sigma_n$ applied to rock samples under direct shear tests. In this research, a two-dimensional particle flow code (PFC2D) is adopted to build a synthetic sandstone rock model with a standard joint roughness coefficient (JRC) profile. The manufactured rock model, which is adjusted by the experiment data and tested by the empirical Barton's shear strength criterion, is then used to research scale effects on the shear strength of rock joints caused by normal stresses. It is found that the failure type can be affected by JRC and $\sigma_n$. Therefore, a scale effect index (SEI) that is equal to JRC plus two times $\sigma_n$ (MPa) is proposed to identify the types of shear failure. Overall, shearing off asperities is the main failure mechanism for rock samples with SEI > 14, which leads to negative scale effects. It is also found that the degree of scale effects on the shear strength of rock joints is more obvious at low normal stress conditions, where $\sigma_n$ < 2 MPa.

**Keywords:** rock joints; shear strength; scale effects; normal stress; JRC; PFC simulation

## 1. Introduction

Rock joints play an important role in the estimation of the shear strength of rock masses [1–3]. Effective design of rock engineering projects, such as underground excavations and open pit slopes, requires precise estimation of the shear strength of rock joints [4–6]. However, it is well known that there is a scale effect on the shear strength of rock joints [7–9]. The main difficulty in determining how the shear strength of rock joints varies with scale is conducting expensive and time-consuming engineering scale in situ testing [10]. Although laboratory scale tests on a small jointed sample cannot generate the precise shear strength of rock joints, they can still reveal the mechanical behavior of jointed rock masses [11]. Therefore, laboratory tests are widely used by researchers to investigate how the shear strength of rock joints is affected by sample sizes. Table 1 presents a review of scale effects on the shear strength of rock joints, which shows conflicting results. The majority of results show that there is a negative scale effect on the shear strength, which means the shear strength decreases with the increase of joint sizes. Some results [12,13] show positive scale effects, which represent the shear strength increases when the joint size increases. While other results [13–15] show no scale effects.

Scale effects on the shear strength of rock joints could be explained in different ways. One explanation is that scale effects occur due to the contact area of joints changing with the increase in joint size [16]. Pratt et al. [14] and Yoshinaka et al. [17] attributed the decrease in shear strength to the smaller contact area of the sample where higher stress was concentrated on these contact surfaces. The other explanation is that the scale effect is associated with the change of undulations and asperities on a joint surface as joint

size increases. A longer sample will result in higher undulation amplitude compared to a smaller sample [18]. Barton and Choubey [19] concluded that the shear behavior of larger rock samples is governed by larger and gentler asperities, while the steep and small asperities are the controlling mechanism in smaller rock samples. Giani et al. [20] stated that when the rock joint shear strength depends on the random distribution of asperity, it will produce a positive scale effect. If the rock joint shear strength depends on wavy and rough surfaces, then there is a negative scale effect. Therefore, more research is required to determine the exact nature of the scale effect on the shear strength of joints.

**Table 1.** Review of scale effects on the shear strength of rock joints [21].

| Authors | Rock Types | Sample Size | Normal Stress (MPa) | Scale Effect |
|---|---|---|---|---|
| Azinfar et al. [13] | Silicon rubber | 25–2500 cm$^2$ | 0.3, 0.8, 1.4 | O, N, P |
| Barton and Choubey [19] | Granite | $9.8 \times 4.5, 45 \times 50$ cm | 0.1–2 | N |
| Bandis et al. [22] | plaster | 6–36 cm | 1 | N |
| Bahaaddini et al. [21] | Sandstone | 5–40 cm | 0.5 | N |
| Castelli et al. [23] | Cement | 100–400 cm$^2$ | 0.75, 1.5, 3 | N |
| Fardin [24] | Concrete | $5 \times 5$–$20 \times 20$ cm$^2$ | 1, 2.5, 5, 10 | N |
| Hencher et al. [25] | Limestone | 44–531 cm$^2$ | 0.0245 | O |
| Johansson [26] | Granite | 36, 400 cm$^2$ | 1 | O |
| Ohnishi et al. [12] | Concrete | 100–1000 cm$^2$ | 0.26–2.04 | P |
| Pratt et al. [14] | Quartz diorite | 60, 142–5130 cm$^2$ | 3 | N |
| Ueng et al. [15] | Cement | 7.5–30 cm$^2$ | 0.3, 0.6, 0.9 | O, N |
| Vallier et al. [27] | - | 10–200 cm | 2 | N |
| Yoshinaka et al. [17] | Granite | 20–9600 cm$^2$ | 0.26–2.04 | N |

"N" means negative scale effect; "P" means positive scale effect; "O" means no scale effect.

Based on the literature review, we noticed that the existing laboratory tests were carried out under various normal stress conditions ranging from 0.0245 to 10 MPa, as shown in Table 1. As we know, the failure mode of rock joints during the direct shear test will be affected by the application of normal stresses. When rock samples are under high normal stress conditions, the tips of asperities could be sheared off; therefore, the shear strength would be relatively higher compared to rock samples that are under low normal stress conditions where sliding is the controlling mechanism of rock failure.

Numerical simulations using the PFC are capable of simulating the asperity damage and degradation process during the shearing tests [28]. It has been proven that the shear strength results acquired from PFC modeling are typically comparable with experimental test results [29]. Therefore, PFC simulations as an alternative to physical testing can be used to reveal the fundamental mechanism of shear behavior of rock joints at various scales.

In this research, a synthetic rock model based on the two-dimensional particle flow code (PFC2D)-based synthetic sandstone rock model is used to study the influence of normal stress on scale effects on the shear strength of rock samples with standard joint roughness coefficient (JRC) profiles, and attempts to answer two questions: (1) Are scale effects on shear behavior affected by normal stresses? (2) What is the degree of scale effects affected by normal stresses?

In this paper, the synthetic rock model for numerical tests is introduced in Section 2. The verification of the synthetic rock model is shown in Section 3. Scale effect investigations are presented and discussed in Sections 4 and 5.

## 2. Synthetic Rock Model for Numerical Tests

### 2.1. Synthetic Rock Model Based on PFC2D

PFC2D is a discrete element program. The bonded particle model (BPM), a composite of rounded particles, simulates complete rock and does not require a continuum-scale constitutive model to depict the mechanical behavior of intact rock [30]. The parallel bond model, which can replicate the physical behavior of a substance similar to cement

linking the two nearby particles, is one of the most fundamental and often used BPMs in the PFC2D, as illustrated in Figure 1. Bond breaking reduces stiffness because contact and bond stiffness both contribute to stiffness in a parallel bond model. While contact stiffness is active as long as particles are in contact, bond stiffness is instantly gone when a bond breaks [31]. Therefore, the parallel bond model is a more accurate bond model for materials that resemble rocks, since it allows for the possibility of bonds breaking in tension or shearing with a corresponding loss in stiffness.

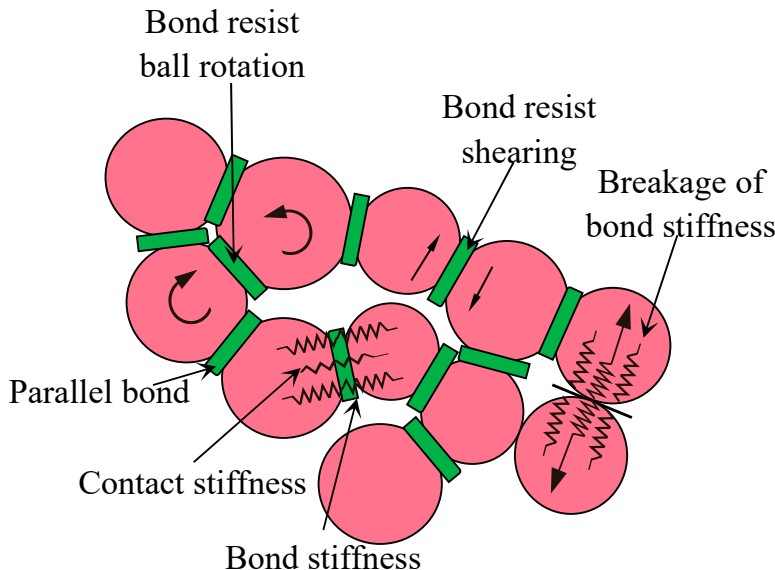

**Figure 1.** Illustration of the parallel bond model.

On the other hand, by adding joints to a BPM assembly using the smooth joint model (SJM), jointed rock masses can be created. The BPM's original contact microscopic characteristics will be replaced with SJM properties with the names friction coefficient $\mu_j$, shear stiffness $k_{sj}$, and normal stiffness $k_{nj}$ when the SJM is put into the BPM [32]. The synthetic rock model constructed by the BPM and SJM has the ability to simulate various mechanical responses of jointed rock masses including peak strength [31], scale effect [33], anisotropy [34], and cracking processes [30,33] in rocks and rock-like materials.

The numerical direct shear test used in this research is presented in Figure 2. The specimen (40 mm × 100 mm) is generated using the BPM. The rock joint is created using the SJM. During the direct shear test, the upper block receives the normal force in a vertical direction. The upper block is given a horizontal velocity of 0.03 m/s, while the lower block is held in place.

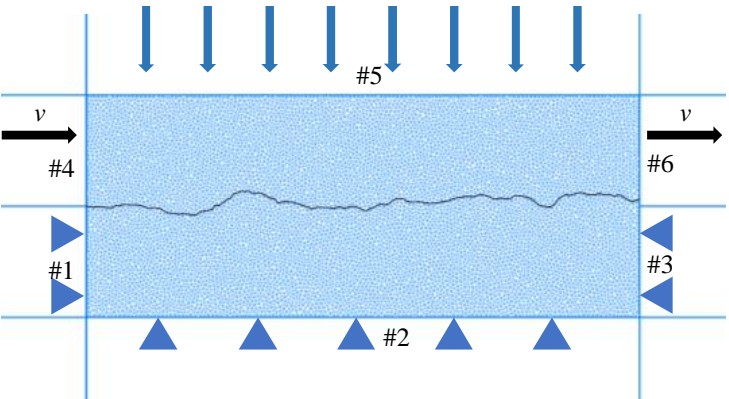

**Figure 2.** Numerical direct shear test set up.

### 2.2. Calibration of Numerical Models

In this study, the synthetic rock model was calibrated using laboratory data from Australia's Hawkesbury Sandstone [28]. Firstly, tests for uniaxial compression on rock samples (42 mm × 84 mm) with a loading rate of 0.02 m/s were performed to determine the BPM's parameters after a calibration process [31] to ensure that the mechanical properties of the synthetic rock model are close to laboratory data.

It should be mentioned that one of the key factors influencing the resilience of restricted materials to deformation and strength, such as rocks and cemented soil, is the loading rate [35–38]. In this research, we did not consider such loading rate effects on the mechanical properties of jointed rocks.

Using the calibrated BPM parameters indicated in Table 2 to perform the uniaxial compression test (Figure 3), the values of elastic modulus $E$, Poisson's ratio $v$, and UCS produced are comparable to experimental tests, as shown in Table 3.

**Table 2.** Micro-parameters of the BPM model.

| Parameters | Values |
|---|---|
| Minimum particle radius: $R_{\min}$ (mm) | 0.28 |
| Maximum particle radius: $R_{\max}$ (mm) | 0.42 |
| Stiffness ratio: $k_{\mathrm{n}} / k_{\mathrm{s}}$ | 2.1 |
| Effective modulus: $E_{\mathrm{c}}$ (GPa) | 4.1 |
| Bond tensile strength: $Tb$ (MPa) | 11.2 |
| Bond friction angle: $\Phi b$ (°) | 35 |
| Cohesion: $cb$ (MPa) | 11.2 |
| Friction coefficient: $u$ | 0.2 |
| Porosity ratio: $e$ | 0.16 |

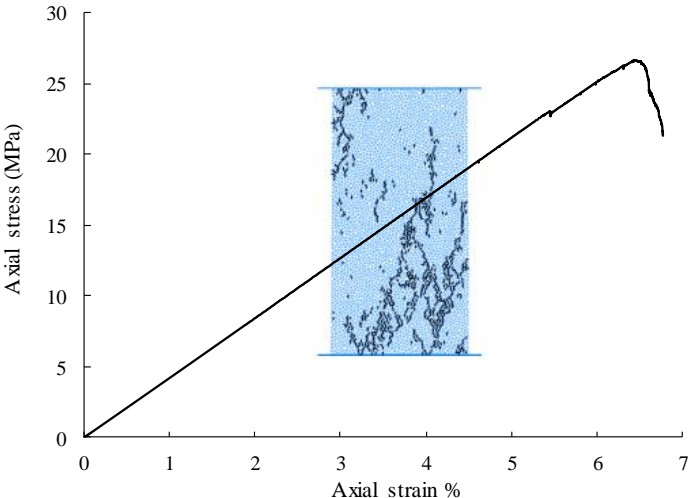

**Figure 3.** Numerical uniaxial compressive test.

**Table 3.** Comparison of mechanical properties calculated from the numerical model and tested from laboratory.

| Properties | Parameters | Laboratory Test | PFC Model |
|---|---|---|---|
| Intact rock properties | UCS (MPa) | 27.40 | 27.40 |
| | $E$ (GPa) | 4.20 | 4.20 |
| | $v$ | 0.20 | 0.21 |
| Joint properties | $K_{\mathrm{n}}$ (GPa/m) | 28.6 | 28.6 |
| | $K_{\mathrm{s}}$ (GPa/m) | 6.40 | 6.40 |
| | $\varphi_{\mathrm{b}}$ (°) | 37.60 | 36.10 |

Then, synthetic rock models (40 mm × 100 mm) with planar joints were constructed. The SJM has the following micro-parameters: friction coefficient $\mu_j$, shear stiffness $k_{sj}$, and normal stiffness $k_{nj}$. In this research, the values of $k_{nj}$ = 25 GPa, $k_{sj}$ = 13 GPa, and $\mu_j$ = 0.75 were selected using the inverse-modeling calibration approach to ensure that the numerical rock model can give a similar response as that from laboratory tests with joint shear stiffness $K_s$ = 6.4 GPa/m, normal stiffness $K_n$ = 28.6 GPa/m, and joint friction angle $\varphi_b$ = 37.6°. The calibration procedure was as follows: (1) The normal deformability compression test was carried out to calibrate normal stiffness $k_{nj}$. (2) The shear test was carried out to calibrate shear stiffness $k_{sj}$ under normal stress of 1 MPa condition. (3) Direct shear tests were undertaken and friction coefficient $\mu_j$ was calibrated. Figures 2–4 present the final mechanical responses of the synthetic rock models after the final calibration.

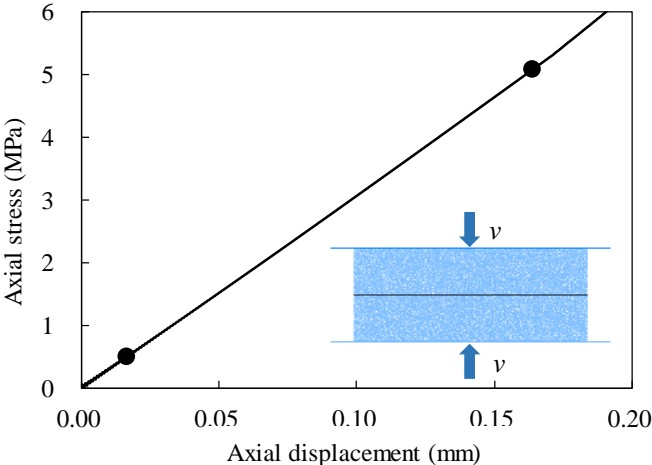

**Figure 4.** The normal deformability test on the synthetic rock specimen with a planar joint.

Figure 4 shows the axial stress-displacement curves of the synthetic rock specimen (40 mm × 100 mm) under the normal deformability test with the loading rate of 0.02 m/s. The value of joint normal stiffness $K_n$ generated by the synthetic rock specimen is 28.6 GPa/m, which is close to the laboratory test results with $K_n$ = 28.8 GPa/m.

Figure 5 shows the shear stress-displacement curve of the synthetic rock model (40 mm × 100 mm) with a planar joint under a direct shear test (loading rate of 0.03 m/s) with the normal stress of 1 MPa. The value of joint shear stiffness $K_s$ generated by the synthetic rock specimen is 6.4 GPa/m, which is the same as laboratory test results with $K_s$ = 6.4 GPa/m.

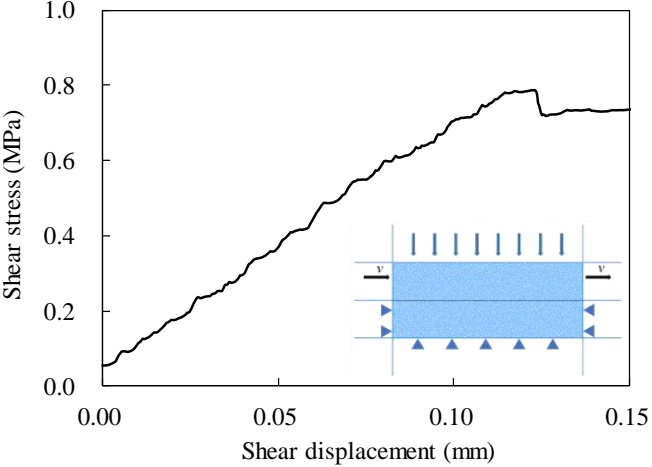

**Figure 5.** The direct shear test on the synthetic rock specimen with a planar joint under normal stress of 1 MPa.

Figure 6 shows the failure envelope of the synthetic rock model (40 mm × 100 mm) with a planar joint under direct shear tests. The value of joint friction angle $\varphi_b$ generated by the synthetic rock specimen is 36.1°, which is close to laboratory test results with $\varphi_b$ = 37.6°.

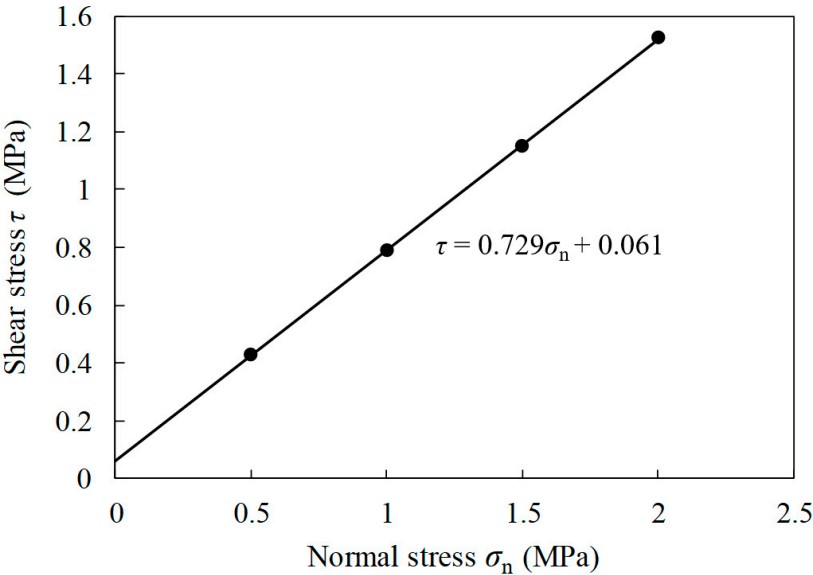

**Figure 6.** Shear strength of the synthetic rock samples under various normal stresses.

## 3. Validation of Synthetic Rock Models

To confirm the reliability of the synthetic rock model shown in Section 2, direct shear tests on the synthetic rock models with 10 standard JRC profiles were performed and the shear strength values produced from numerical simulations were compared to those derived from Barton's empirical shear strength model.

### 3.1. Barton's Shear Strength Model

One of the most widely adopted empirical strength criteria for estimating rock joint shear strength in rock engineering is the Barton's shear strength criterion. Based on the results of a large number of shearing tests on various rock joint profiles, Barton and his co-workers [19,39] proposed an empirical equation to estimate the shear strength of rock joints, as shown in Equation (1).

$$\tau = \sigma_n \tan \left[ \varphi_b + \text{JRC} \, \lg \left( \frac{\text{JCS}}{\sigma_n} \right) \right] \tag{1}$$

where $\varphi_b$ is the joint friction angle. JCS is the joint compression strength, which is equal to UCS of intact rock in this research. JRC stands for joint roughness coefficient and can be calculated using standard joint profiles.

### 3.2. Numerical Simulation Results

We performed extensive numerical direct shear experiments on synthetic rock models with varied JRC profiles in normal stress levels between 0.5 and 5 MPa. The failure envelopes generated by direct shear tests on synthetic rock models were compared to the empirical Barton's shear strength criterion (Equation (1)) with $\varphi_b$ = 36.1° and JCS = 27.4 MPa. Figure 7 compares the shear strength obtained from numerical simulations to Barton's model, indicating that the usage of synthetic rock models is capable of generating adequate shear strength of rock joints.

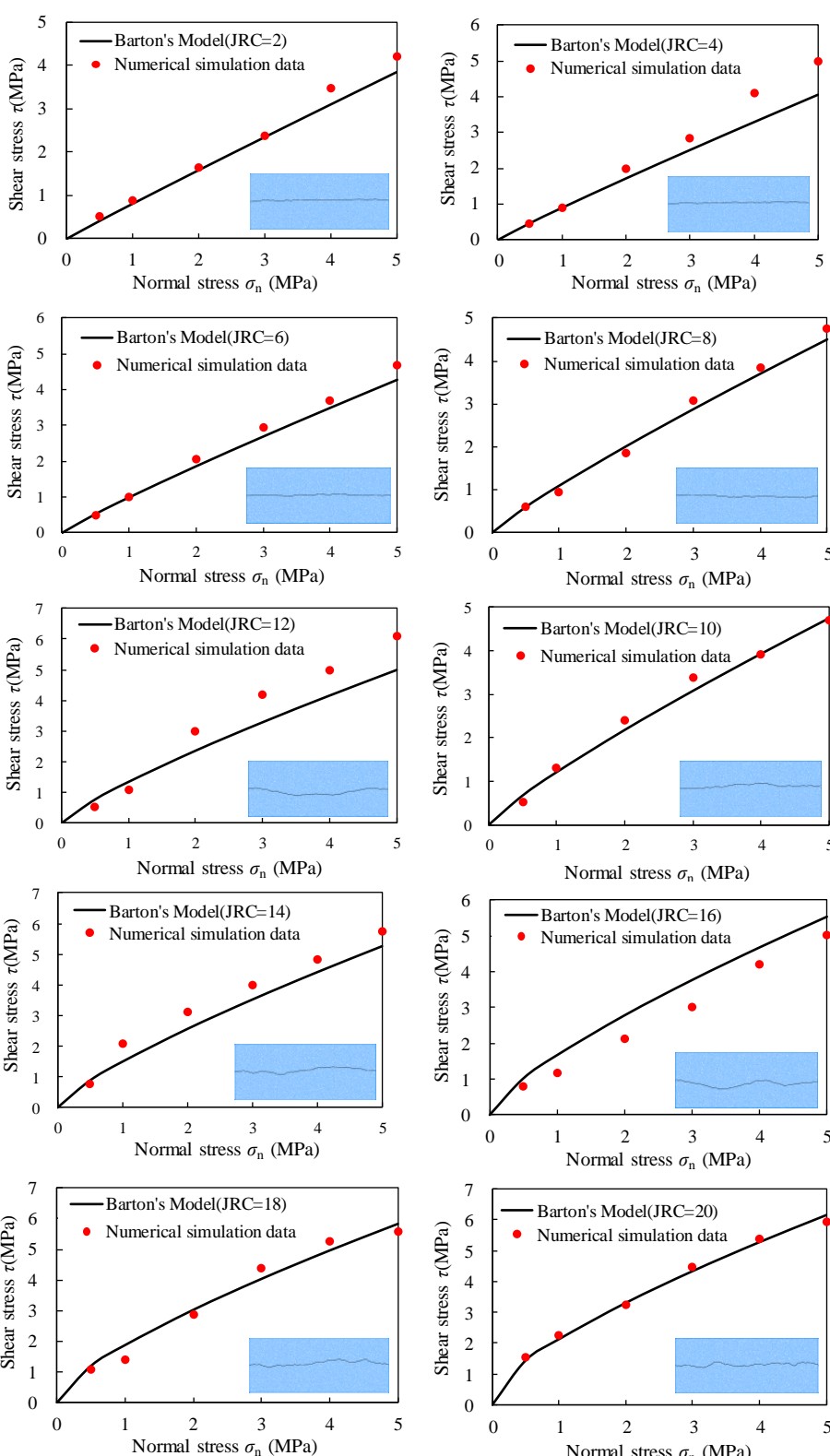

**Figure 7.** Comparison failure envelopes obtained from numerical simulations and the Barton's empirical model.

## 4. Configuration of Rock Samples for Scale Effect Investigations

Two methods are widely used for investigating scale effects on the shear strength of rock joints [15]. The first one is to divide a large rock joint into several smaller sizes of rock joints, as shown in Figure 8a, which presents an example of the division of the Barton's

typical profile. The geometry characteristics or the values of JRC of smaller sizes of rock joints can be different from that of the original larger rock joint. The other method is the assembly of several repeated 100 mm profiles into larger rock joints many times the original profile length, as shown in Figure 8b. The joint roughness or the value of JRC of assembly samples is the same as that of the original joint surface.

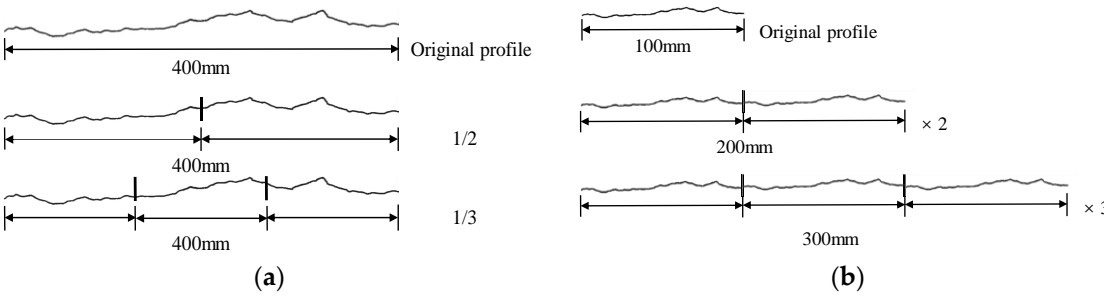

**Figure 8.** Two types of configuration of samples for scale effect investigations (**a**) division of the Barton's JRC profile (**b**) assembly of repeated Barton's JRC profile.

In fact, the scale effect on the shear strength of rock joints includes two factors, which are the sample size itself and the geometrical characteristics of the joint surface. Rock samples generated by the division method have various sample sizes and geometry characteristics. However, the rock joints generated by the assembly method have various sample sizes but have the same geometry characteristics. It is well known that the geometry characteristics will affect the shear strength of rock joints [8]. Therefore, in this research, we adopt the assembly of a repeated model that has the same geometry characteristics and JRC values to research the influence of pure sample size on the shear strength of rock joints.

## 5. Results and Discussion

Once synthetic rock models were validated, a series of rock specimens with various JRC profiles and different sizes (40 mm × 100 mm, 80 mm × 200 mm, and 120 mm × 300 mm) were generated to study the effect of sample sizes on rock joint shear strength. The shear strength values of different sizes of rock samples under given normal stresses were calculated and are summarized in Figure 9.

In this research, the index $k$ (see Equation (2)), which is the average slope of three points, was used to identify the types of scale effects.

$$k = \frac{N\sum x_i y_i - \sum x_i \sum y_i}{N\left(\sum x_i^2\right) - \left(\sum x_i\right)^2} \tag{2}$$

where $x_i$ is the joint length of the rock sample, $y_i$ the shear stress of the rock sample, and $N$ is the number of the testing sample. $k > 0$ means the rock joint has a positive scale effect and $k < 0$ means the rock joint has a negative scale effect. The value of $k$ can be calculated using three groups of data. For example, for rock samples with JRC = 2 under the normal stress $\sigma_n = 5$ MPa, the shear strength of rock samples with joint lengths $l = 100$, 200, and 300 mm are 4.2, 4.4, and 4.6 MPa, respectively. Therefore, data (100, 4.2), (200, 4.4), and (300, 4.6) were put into Equation (2) to calculate the value of $k$. The result shows $k = 0.4$, which means the scale effect is positive. Table 4 shows comprehensive scale effect results of rock samples with various JRC profiles under different normal stress conditions. In Table 4, P means positive scale effect and N means negative scale effect.

The results presented in Table 4 are also plotted in Figure 10. It is found that the failure mode of rock joints during the direct shear test will be affected by the applying normal stresses and the joint roughness. When rock samples under high normal stress conditions, the tips of asperities with large joint roughness coefficient could be sheared off, therefore, the number of shear crack is relatively higher compared with rock samples with small joint roughness coefficients under low normal stress conditions where sliding is

the controlling mechanism of rock failure. For example, when a rock sample with JRC = 2 under the normal stress $\sigma_n$ = 0.5 MPa, the number of shear cracks is 10 and the scale effect is positive. However, when a rock sample with JRC = 20 under the normal stress $\sigma_n$ = 5 MPa, the number of shear cracks is 280 and the scale effect is negative.

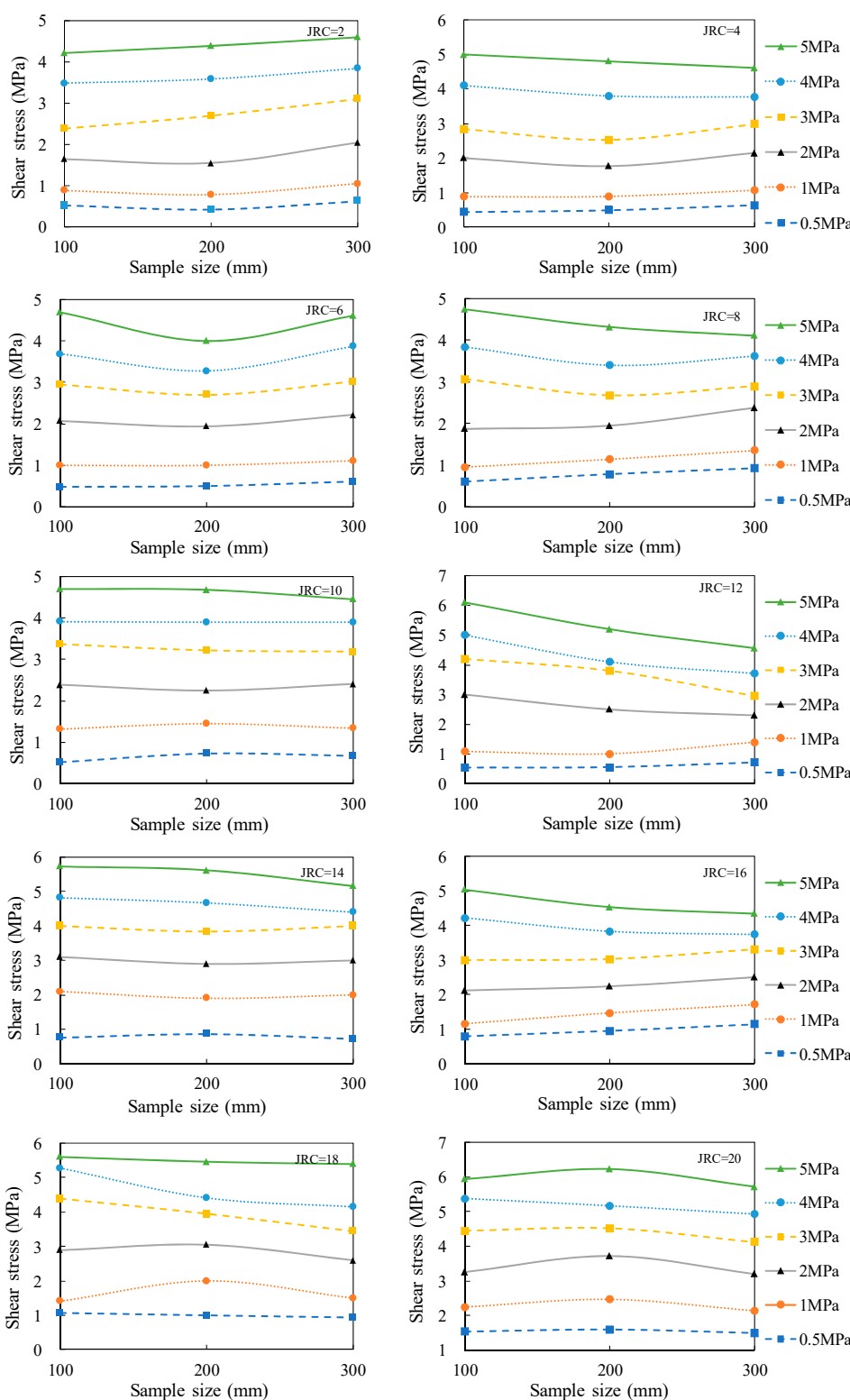

**Figure 9.** Results of scale effects on the shear strength of rock samples under various normal stress conditions.

**Table 4.** Results of scale effects of shear strength of rock joints.

| JRC | Normal Stress $\sigma_n$ (MPa) | | | | | |
|---|---|---|---|---|---|---|
| | 0.5 | 1 | 2 | 3 | 4 | 5 |
| 2 | P3 | P4 | P6 | P8 | P10 | P12 |
| 4 | P5 | P6 | P8 | P10 | N12 | N14 |
| 6 | P7 | P8 | P10 | P12 | P14 | N16 |
| 8 | P9 | P10 | P12 | N14 | N16 | N18 |
| 10 | P11 | P12 | P14 | N16 | N18 | N20 |
| 12 | P13 | P14 | N16 | N18 | N20 | N22 |
| 14 | N15 | N16 | N18 | N20 | N22 | N24 |
| 16 | P17 | P18 | P20 | P22 | N24 | N26 |
| 18 | N19 | P20 | N22 | N24 | N26 | N28 |
| 20 | N21 | N22 | N24 | N26 | N28 | N30 |

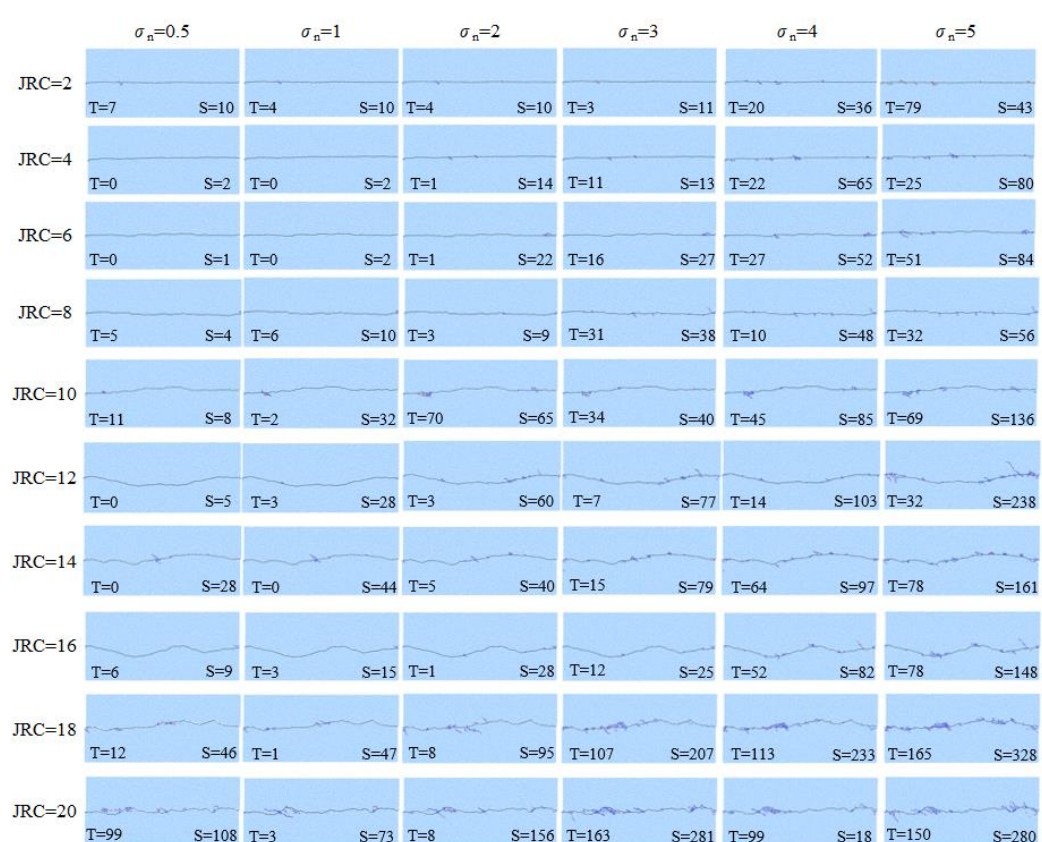

**Figure 10.** Failure pattern and crack number of rock samples (40 mm × 100 mm) at peak shear strength.

Based on the results of Table 4, a scale effect index (*SEI*) which is equal to *JRC* plus two times of normal stress (MPa), as shown in Equation (3), was proposed to identify the types of scale effects.

$$SEI = JRC + 2\sigma_n \tag{3}$$

The values of *SEI* for rock samples under different normal stress conditions are given in Table 4. For example, for rock samples with *JRC* = 2 under the normal stress $\sigma_n$ = 0.5 MPa, the value of *SEI* = 2 + 2 × 0.5 = 3. The number P3 in Table 4 means the rock sample with *SEI* = 3 has a positive scale effect. It was found that 20 out of 21 rock samples have negative scale effects when *SEI* > 14, and 29 out of 33 rock samples have positive scale effects when *SEI* < 14.

To find the possible reason why the use of SEI can identify the types of scale effects on shear strength, we monitored the crack number generated in the synthetic rock sample

when the stress reaches the peak strength during the direct shear tests. When the parallel link between nearby particles in the PFC rock model is broken, a micro-tensile crack or micro-shear crack can occur. Figure 10 shows the failure pattern corresponding to the shear crack number of each rock sample when the shear stress reaches the peak strength. For example, the S = 10 represents a shear crack number of 10 and T = 7 represents a tension crack number of 7 for a sample with SEI = 3 (JRC = 2 and $\sigma_n$ = 0.5 MPa). The relations between SEI values and shear crack numbers of rock samples are also plotted in Figure 11. We can find that the number of shear cracks is low when SEI < 14. However, the number of shear cracks dramatically increases when the value of SEI is over 14, where the controlling failure mechanism transforms sliding to shearing off asperities.

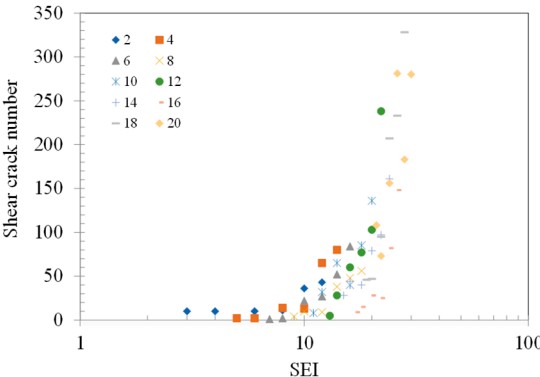

**Figure 11.** Relations between SEI and shear crack number of rock samples.

Therefore, we can conclude that the results presented in Table 4 and Figure 11 show that the proposed SEI is capable of identifying types of scale effects. When SEI < 14, sliding over joints is the controlling mechanism of rock failure, which leads to positive scale effects; however, shearing off asperities could be the controlling mechanism of rock failure for rock samples with SEI > 14, which leads to negative scale effects.

On the other hand, to further identify the degree of scale effects, the coefficient of variance (CV), which can calculate the value of Standard Deviation/Mean to quantify the random influence of a bunch of data, was further used as a scale effect magnitude index to quantify scale effects on shear strength of rock joints caused by normal stress conditions. The calculation results are presented in Figure 12.

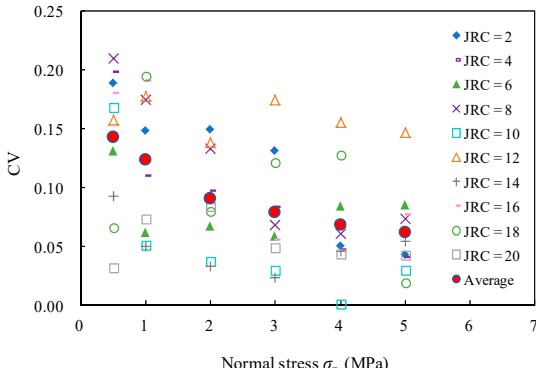

**Figure 12.** The degree of scale effect caused by normal stresses.

It is found that, in general, the values of CV decrease with the increase of normal stress. We also calculated the average value of CV for a given group of data. Figure 12 shows that the values of average CV decrease when the normal stress increases from 0.5 to 2.0 MPa, then, it tends to be stable with further increase of normal stresses, which means the degree of scale effects on shear strength of rock joints is more obvious at low normal stress conditions where $\sigma_n$ < 2 MPa.

Such a phenomenon can also be validated by the laboratory data published by Fardin [24], who carried out a laboratory study of the scale effect on the shear strength of concrete replicas with roughness joints. Laboratory test results are shown in Figure 13. The CV values of samples under a specific normal stress condition were calculated and are shown in Figure 14. The value of CV is up to 0.4 when $\sigma_n$ = 1 MPa, then, it decreases sharply to 0.14 when $\sigma_n$ increases to 2 MPa. After that, there is a slight change in CV values with a further increase of $\sigma_n$ from 2 MPa to 10 MPa. Such change in CV values with normal stresses is similar to that of the numerical results in Figure 12.

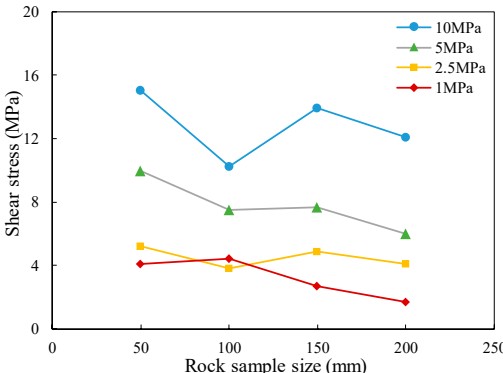

**Figure 13.** Laboratory data of scale effect on shear stress of rock joints under various normal stress conditions (data from Fardin [24]).

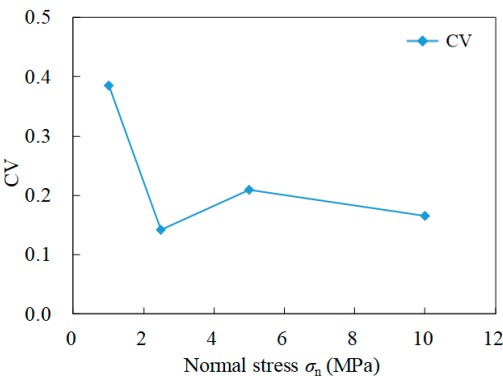

**Figure 14.** Laboratory investigation on the degree of scale effect caused by normal stresses.

## 6. Conclusions

Synthetic rock models with standard JRC profiles were constructed in PFC2D to investigate scale effects on the shear strength of rock joints under various normal stress conditions. The capability of the synthetic rock model to simulate the shear behavior of rock joints was tested by comparing numerical simulations with the Barton's shear strength criterion.

Once synthetic rock models were validated, a series of rock specimens of different sizes (40 mm × 100 mm, 80 mm × 200 mm, and 120 mm × 300 mm) were generated to investigate the influence of sample sizes on rock joint shear strength under normal stress ranges from 0.5 to 5 MPa.

Numerical simulation results show that the types of scale effects could be affected by the JRC profiles and normal stresses. Therefore, a scale effect index (SEI) that is equal to JRC plus two times normal stress (MPa), as shown in Equation (3), was proposed to identify the types of scale effects. It is found that for the rock sample with SEI < 14, sliding over joints is the controlling mechanism of rock failure, which leads to positive scale effects. However, shearing off asperities could be the controlling mechanism of rock failure for rock samples with SEI > 14, which leads to negative scale effects.

We also further investigated the influence of normal stress on the degree of scale effects on the shear strength of rock joints. It is discovered that the degree of scale effect is more obvious at low normal stresses conditions where $\sigma_n < 2$ MPa.

Finally, it should be noted that the finding of this research is based on the analysis of test data of Australia Hawkesbury Sandstone. Therefore, the finding of this research is open for further improvements as more shear strength data of various rock types become available.

**Author Contributions:** Conceptualization, J.S.; methodology, C.S. and H.H.; software, J.S. and H.H.; validation, J.S. and H.H.; formal analysis, J.S. and H.H.; investigation, J.S. and C.S.; resources, J.S. and C.W.; data curation, C.S. and H.H.; writing—original draft preparation, J.S. and H.H.; writing—review and editing, J.S., C.S., J.C. and C.W.; visualization, C.S. and H.H.; supervision, J.S. and C.W. All authors have read and agreed to the published version of the manuscript.

**Funding:** This research was funded by the Zhoushan Science and Technology Bureau (2022C81001), the Key R&D of Zhejiang Province (2021C03183), and the Scientific Research Fund of Zhejiang University (XY2021011), the Sanya Yazhou Bay Science and Technology City (KYC-2020-01-001), the Finance Science and Technology Project of Hainan Province (ZDKJ202019).

**Institutional Review Board Statement:** Not applicable.

**Informed Consent Statement:** Not applicable.

**Data Availability Statement:** The data presented in this study are available on request from the corresponding author.

**Acknowledgments:** We express our sincere gratitude to the reviewers for their valuable comments and suggestions.

**Conflicts of Interest:** The authors declare no conflict of interest.

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
