# Peer review of "Scale Effects on Shear Strength of Rough Rock Joints Caused by Normal Stress Conditions"

_sustainability, doi:10.3390/su15097520_

Round 1

Reviewer 1 Report

The authors presented numerical investigation to study the influence of normal stress on scale effects on shear strength of rock samples. They found that the failure type could be affected by joint roughness and applied normal stress conditions. This reviewer found the manuscript well written, the analysis well developed and described, and language is also good. I recommend to accept the article, some minor revise as follows:

1. In this research, the authors adopt the assembly of repeated model which has the same geometry characteristics and JRC values to investigate the influence of sample size on the shear strength of rock joints, which should be stated in the manuscript to let the readers know the limitation of this research.

2. It is known that there are two types of cracks in a PFC model, namely, shear crack and tensile crack, therefore, what types of crack in Fig. 8 ?

3. The joint profile was simulated in PFC using the point spacing of 1 mm. does means the asperities smaller than 1 mm can not be simulated in numerical models ?

4. Replace the x and y with sig_n and tau of formula in Fig. 4.

5. Page 6, "is same as" should be "is the same as"

6. Table 3 , " Kn  (GPa/m)" should be " Kn (GPa/m)".

7. Table 3 , " φb  (°)" should be " φb (°)".

Reviewer 2 Report

Title: Scale Effects on Shear Strength of Rough Rock Joints Caused by Normal Stress Conditions

The paper presents a numerical study on the scale effects on the shear strength of rough rock joints caused by normal stress conditions by using two-dimensional particle flow code (PFC2D) based synthetic sandstone rock models. The scientific writing of this paper is satisfactory. The results are generally presented in a systematic manner, but some of the details need more explanation. Based on a thorough review of the manuscript, the reviewer suggests that the paper requires "Major Revision". Following are the detailed comments of the reviewer:

Major Comments:

1.       Section-2 Synthetic rock model for numerical tests: Authors should add more details of bonded particle model (BPM) for the readers, and should also include the references of the published studies to justify the use of this model.

2.       Section-2 Synthetic rock model for numerical tests: “… axial stress-displacement curves of the synthetic rock specimen 124 (40 mm × 100 mm) under the normal deformability test with the loading rate of 0.02m/s”. Loading rate is one of the crucial parameter which affects the strength and deformation response of the bounded materials such as rocks and cemented soil. The selection of 0.02 m/s should properly by justified as the strength characteristics of the simulated synthetic rock is expected to be affected largely by changing the loading rate. Authors should add some literature on this aspect as the results presented in the study did not consider the effects of viscosity (loading rate). The reviewer also suggest to include the findings of other studies related to loading rate dependency of different geomaterials for instance:

Maqsood, Z., Koseki, J., Miyashita, Y., Xie, J., & Kyokawa, H. (2020). Experimental study on the mechanical behaviour of bounded geomaterials under creep and cyclic loading considering effects of instantaneous strain rates. Engineering Geology, 276, 105774.

G. Swan, J. Cook, S. Bruce, R. Meehan, Strain rate effects in Kimmeridge Bay shale, Int. J. Rock Mech. Mining Sci. Geomech. Abstracts 26 (2) (1989) 135–149.

Maqsood, Z., Koseki, J., Ahsan, M. K., Shaikh, M., & Kyokawa, H. (2020). Experimental study on hardening characteristics and loading rate dependent mechanical behaviour of gypsum mixed sand. Construction and Building Materials, 262, 119992.

3.       Section 2.2 Calibration of numerical models: “In this research, the laboratory data of Australia Hawkesbury Sandstone [28] was used to calibrate the synthetic rock model.” Please justify the reason for the selection of Australia Hawkesbury Sandstone for the calibration of the synthetic rock model used in this study. The authors should also discuss the applicability of the findings of this study for other types of rocks. The reviewer suggests to at least consider another set of laboratory data of different type of rock to confirm whether similar results could be obtained or not.  

4.       Fig. 9: The results presented in Fig. 9 are relatively scattered. Authors should comprehensively discuss this aspect. Also, the authors should try to change the vertical scale from normal to the logarithmic scale to more explicitly show the trend.

Minor Comments:

1.       Table-1: Please use superscripts for units mentioned in this table.

Based on these comments, the reviewer suggests that the paper requires "Major Revision".

Reviewer 3 Report

In this paper, the authors investigated the scale effects on the shear strength of rough rock joints by DEM simulation. The results are interesting and can promote application prospects. Relevant conclusions have been obtained. It is recommended to be accepted for publication pending suitable minor revision.

1 The numerical results show the rock sample with SEI < 14, sliding over joints is the controlling mechanism of rock failure, which leads to positive scale effects. It is recommended to cite some recently published research on scale effects.

2 Please explain the meaning of the numbers in the photo in Fig.8.

3 Please indicate how many particles were used in the model and show at least as a graph rather than a table in the validation session.

4 In line 200 of p8, the authors' interpretation of equation 2 is incorrect, please check.

5 It is recommended that the corresponding shear and tension cracks be calibrated in Figure 8 so that the failure mechanism can be better analyzed.

6 Some of the literature is very old and it is recommended to add some relevant studies published in recent years.

Reviewer 4 Report

The paper is interesting and clearly has merits. It examines scale effects on shear strength in rocks joints under normal stress and draws useful conclusions.

However, the following points should also be addressed by the authors:

     1.       The manuscript has a few minor English mistakes that can be directly addressed by the authors.

      2.       The sentence in the abstract in L.13-16 is too long. It should be reviewed and rewritten using multiple sentences.

     3.       In L. 17: “It is found that the failure type could be affected by JCR and sigma_n”. The reviewer suggests to avoid the use of “could” and to be more specific. The same for the sentence in L.19.

     4.       The authors should clearly indicate at the end of the introduction the outline of the rest of the paper, section by section, as is usually done in scientific articles.

    5.       The authors should clearly state at the end of the introduction the objectives of the paper, as is usually done in scientific articles.

    6.       The reviewer suggest moving the two paragraphs in L.69-78 to the introduction, so that a few details are given in the introduction on the PFC2D code.

     7.       The sentence in L.90-93, “The synthetic rock model constructed by the BPM and SJM has the ability to simulate various mechanical responses of jointed rock masses including peak strength, scale effect, anisotropy, and cracking processes in rocks and rock-like materials.” should be justified with at least one citation or eliminated.

These aspects can be directly addressed in a minor revision.

Round 2

Reviewer 2 Report

The quality of the manuscript has been improved to a satisfactory level. The authors have effectively responded to all the comments and new details are added in the revised manuscript.